# Effect of Diet Supplementation with Quinoa Seed and/or Linseed on Immune Response, Productivity and Meat Quality in Merinos Derived Lambs

**DOI:** 10.3390/ani8110204

**Published:** 2018-11-10

**Authors:** Rosaria Marino, Mariangela Caroprese, Giovanni Annicchiarico, Francesco Ciampi, Maria Giovanna Ciliberti, Antonella della Malva, Antonella Santillo, Agostino Sevi, Marzia Albenzio

**Affiliations:** 1Department of Agricultural Food and Environmental Sciences, University of Foggia, Via Napoli, 25-71121 Foggia, Italy; mariangela.caroprese@unifg.it (M.C.); francesco.ciampi@unifg.it (F.C.); maria.ciliberti@unifg.it (M.G.C.); antonella.dellamalva@unifg.it (A.d.M.); antonella.santillo@unifg.it (A.S.); agostino.sevi@unifg.it (A.S.); marzia.albenzio@unifg.it (M.A.); 2Council for Agricultural Research and Economics—Research Centre for Animal Production and Aquaculture (CREA-ZA), S.S.7 Via Appia, 85051 Bella Muro (PZ), Italy; giovanni.annicchiarico@crea.gov.it

**Keywords:** lamb, linseed, quinoa, metabolic profile, immune response, meat quality

## Abstract

**Simple Summary:**

Stressful conditions can play an important role in affecting welfare, productive performances and meat quality of lambs. The relation between nutrition and immune response has been investigated in the livestock production, particularly in dairy cattle and sheep. Due to costs related to animal feed it is useful to evaluate the proper feeding strategy supplementation for improving animal welfare and lamb meat quality. The present study aimed to evaluate the effects of supplementation with linseed, quinoa seed and their combination on metabolic profile, immune system, and cortisol response in blood and on meat quality of merinos derived lambs. Both linseed and quinoa supplementation enhanced the immune responses of lambs, and showed a hypo cholesterol effect on blood of lambs. Moreover, lambs supplemented with linseed resulted in the lowest level of cortisol secretion during the loading test demonstrating the link between stress and the immune system. In addition, data from the present experiment highlighted that linseed supplementation in lambs enhance meat quality producing a better meat tenderness. These findings should be considered for development of specific strategies aimed at improving the quality of meat and sustaining lambs’ welfare.

**Abstract:**

In the last years several studies have investigated the strong relation between nutrition and immune response in the livestock production, particularly in dairy cattle and sheep. The aim of this study was to evaluate the effects of supplementation based on linseed, quinoa seeds and their combination on welfare, productivity and quality of meat from merinos derived lambs. 32 weaned lambs were divided into 4 experimental groups: quinoa (Q), linseed (LS) and combination of quinoa and linseed (LS + Q) that received the respective supplementation and control group (C) without supplementation. Lambs from all supplemented groups showed lower plasma urea, creatinine and cholesterol than control. Both linseed and quinoa supplementation enhanced the cell-mediated immune responses of lambs, furthermore, linseed supplementation resulted in the lowest level of cortisol secretion after handling, loading and transport. Meat from lambs supplemented with linseed and LS + Q showed the highest pH, at 1 and 3 h post-mortem, while, meat from all supplemented groups was more tender than meat from control. Results indicated that linseed and quinoa seeds supplementation can help the animal to cope with stressful events due to the close link between stress responses and the immune system and for improving meat quality in terms of better tenderness.

## 1. Introduction

Consumer perception and preference for meat products are affected by many factors that contribute to define meat quality, defining direct factors linked to the muscle, nutritional, sensorial and technologic factors, and some factor related to the image of the product such as the welfare of the animals [1]. Feeding with a proper supplementation can improve lamb meat quality, modifying growth performance carcass characteristics [2,3], and meat quality traits [3,4].

In the last two decades several studies investigated the strong relation between nutrition and immune response [5]. Particularly, the effect of polyunsaturated fatty acids (PUFA) on the immune responses [6], oxidative stress [7], ruminal metabolism [8] has been extensively explored on dairy sheep. Whole linseed supplementation, rich in PUFA, exerted a modulation of immune responses, enhancing humoral responses and restoring thermal homeostasis in dairy sheep during high ambient exposition [9,10]. Recently, quinoa seeds, a pseudo-cereal has been considered as a novel attractive product for animal nutrition, in relation to triterpenoid saponins content and their physiological functionalities, among which anti-microbial [11], anti-inflammatory [12] and cholesterol lowering activities [13]. Furthermore, it has been determined that lipidic fraction of quinoa was also rich in PUFA, particularly in linoleic and linolenic acid [14].

Moreover, stressful conditions such as human–animal relationship can play an important role in affecting welfare, productive performances and meat quality of lambs [15]. To respond to stressful event, ruminants activate an increasing of secretion of hormones such as ACTH, cortisol, adrenalin and noradrenaline into the bloodstream. Cortisol, a glucocorticoid hormone, increase in response to physical and psychological stress; however, this elevation is not totally proportional to the degree of stress being affected also by stress perception, individual emotional reactivity, habituation to stress, and diet [10]. Indeed, glucocorticoids have been classified as immunosuppressive and anti-inflammatory molecules. More recently, it has been found a bidirectional effect of cortisol on the immune responses in relation to the magnitude and the duration of stress [16].

Although the effect of dietary linseed sources on productive performances and lamb meat quality has been studied, little information is available on the effect on welfare parameters, including cortisol production, cellular and humoral immune responses. In addition, the effect of the inclusion of quinoa in ruminant’s diet on animal welfare and on meat quality have not been previously investigated.

Our hypothesis was that PUFA enriched-diet could be able to sustain immune and metabolic responses of lambs, promoting welfare, and, as a consequence, increase the productivity, and the whole quality of meat from merinos derived lambs. Therefore, the aim of this study was to evaluate the effects of supplementation based on linseed, quinoa seeds and their combination on immune responses, productivity and quality of meat from merinos derived lambs.

## 2. Materials and Methods

The experimental design and all animal procedures were developed in accordance with the Foggia University Institutional Animal Care and Use Committee (protocol number 003-2016). The experiment was conducted at the research station of the Council for Agricultural Research and Economics (CREA-ZA) (latitude: 41°27′6′′ and longitude: 15°33′5′′) located in Apulia Region (Segezia, Southern Italy). A total of 32 Italian Merino male lambs after weaning (42 ± 2 days of age), with average initial live weight 14.30 ± 1.22 kg, were used in this experiment. Lambs were randomly allotted to four dietary groups (8 per treatment group): control (C), fed with vetch, oat hay and pelleted concentrate; quinoa (Q), fed with vetch, oat hay and pelleted concentrate and supplemented with quinoa seed; linseed (LS), fed with vetch, oat hay and pelleted concentrate and supplemented with linseed; quinoa and linseed (LS + Q), fed with vetch, oat hay and pelleted concentrate and supplemented with of quinoa seed and linseed. During the 50-day feeding experiment, lambs were housed in individual pens equipped with water dispenser. Ingredients and chemical composition of the diets are shown in Table 1. However, details on the experimental design has been reported by della Malva et al. [17].

### 2.1. Feed Analysis

Analysis on feed were carried out on pooled feed samples collected throughout the trial period. Neutral detergent fiber (NDF) and acid detergent fiber (ADF) were determined according to Van Soest et al. [18] while the AOAC [19] methods were used for crude fat (Ether extract; method 935.38) and crude protein (CP; method 984.13) measurements. Metabolizable energy was calculated using INRA system [20]. Fatty acids composition was determined using the procedure described by O’Fallon et al. [21].

### 2.2. Weight Gains, Post-Slaughter Measurements and Meat Sample Collection

Animals were individually weighed at weekly intervals to estimate the average daily gain (ADG, g/d). At the end of the experiment, animals were fasted for 12 h, weighed and transported to a local slaughterhouse where they were slaughtered, according to industrial routines used in Italy and to the EU rule n. 1099/2009 [22]. Carcasses were chilled at 2–4 °C, weighed and pH was measured at 1, 3, 6 and 24 h using a portable pH-meter equipped with a glass electrode and an automatic temperature compensator (CAT) probe (Hanna Instruments, Woonsocket, RI, USA) inserted in *quadriceps femoris* muscle. After 24 h post-mortem from each half carcass *quadriceps femoris* muscle was removed; chemical composition, Warner Bratzler shear force (WBSF) and colorimetric parameters were evaluated.

### 2.3. Blood Sampling and Metabolic Profile Determination

Blood samples (7 mL) were collected by jugular venipuncture at d 0, 15, 25, 35, 49 (5 recordings) at 7 am using vacutainer tubes containing Sodium–heparin. Blood samples were immediately centrifuged (3000 rpm for 15 min), then plasma samples were collected and immediately frozen at −20 °C until analysis. Plasma metabolites (glucose, cholesterol, urea, total protein, albumin, creatinine and NEFA) and enzymatic activities (alkaline phosphatase, ALP; lactate dehydrogenase, LDH; γ-glutamyl transferase, GGT) were determined using an automated analyzer for biochemical chemistry (ILAB 600 Plus; Instrumentation Laboratory, Lexington, MA, USA), and its dedicated kits. Plasma globulin concentration was obtained as the difference between total protein and albumin.

### 2.4. Establishment of Humoral Response to a Keyhole Limpet Hemocyanin (KLH)

To determine the lambs’ humoral response during the trial, the animals received a subcutaneous injection of the antigen KLH (Sigma Aldrich, Milan, Italy), to which the animals had not previously been exposed. At the start of the experiment, 0 day, 0.5 mg of KLH dissolved in 1 mL of sterile saline solution and 1 mL of incomplete Freund’s adjuvant (Sigma Aldrich, Milan, Italy) were injected subcutaneously [23]. A subsequent injection of 0.5 mg KLH in saline without adjuvant was administered 15 day later. The anti-KLH antibody titers in lamb plasma samples were evaluated by an ELISA performed in 96-well U-bottomed microtiter plates according to Caroprese et al. [24]. Briefly, wells were coated overnight with 100 µL of antigen (0.5 mg of KLH/mL of phosphate-buffered saline-PBS) at 4 °C, then washed (PBS/0.05% Tween 20—PBST) and incubated with 3% of Bovine Serum Albumin (BSA) dissolved in PBST (200 µL) at 37 °C for 1 h to reduce non-specific binding. After washing, the plasma (1:1.000 diluted in PBS; 100 µL per well) was added and incubated at 37 °C for 1 h. The extent of antibody binding was detected using a horseradish peroxidase-conjugated donkey anti-sheep IgG (Sigma Aldrich, Milan, Italy). Optical density was measured at a wavelength of 450 nm using Power Wave XS (Biotek, UK). Plasma samples were read against a standard curve (Y = (0.00152 − 1.18)/(1 + (x/1.59 × 10^3^)^1.15^ + 1.18, R^2^ = 99.8%) obtained using scalar dilution of ovine IgG (Sigma Aldrich, Milan, Italy). Data were expressed as µg/mL. The assay was optimized in our laboratory for dilution of plasma, concentration of coating antigen and detector antibody.

### 2.5. Evaluation of the Cell-Mediated Immune Response and Cortisol Secretion after Loading Test

At 0, 25, and 35 day the in vivo cell-mediated immune response after intradermal injection of the mitogen phytohemoagglutinin (PHA) (1 mg/mL, Sigma Aldrich, Milan, Italy) suspended in sterile saline solution (1 mL) was evaluated on each lamb by measuring lymphocyte proliferation. The lymphocyte proliferation was measured as the changes in skin-fold thickness after 24 h of PHA injection according to Caroprese et al. [23].

At 45 day of the experiment a loading/unloading test in order to simulate stress related to handling, loading, transport and unloading and to evaluate cortisol secretions. Before loading (0 min), after transport and unloading at 10 min, and 60 min after unloading, blood samples were collected into EDTA-K_3_ vacuum tube (Becton Dickinson, Plymouth, UK) to determine the cortisol level. Hormone concentration was determined by colorimetric competitive sheep ELISA (Sigma Aldrich, Milan, Italy) according to the manufacturer’s instruction. The sensibility of the assay was 0.1 ng/mL. Data were expressed as ng/mL of cortisol.

### 2.6. Meat Quality: Chemical Composition, Color and Instrumental Tenderness of Meat

Meat sample was ground to homogeneous consistency using a food processor before chemical analysis. Moisture, protein, fat and ash content were determined according to methods recommended by the AOAC [19].

Color was measured using a color meter Minolta CR 400 (Konica Minolta, Osaka, Japan) on 1 cm thick steaks after storage at 3 ± 1 °C for 1 h. Color measurements were expressed as L* (lightness), a* (redness) and b* (yellowness) according to the standard conditions of the Commission International d’Eclairage (CIE). Results were reported as the average of five measurements taken from random locations on the surface of each slice.

The instrumental tenderness was estimated by using Warner–Bratzler shear force (WBSF) test on cooked meat as described by Marino et al. [25]. Measurements were conducted using an Instron 3343 universal testing machine with a 500 N load cell (Instron Ltd., High Wycombe, UK) fitted with a Warner–Bratzler blade. For each sample, 6 replicates were performed and the mean of all the replicates was used for statistical analysis.

### 2.7. Statistical Analysis

Data of immunological parameters, metabolic profile and cortisol concentrations were checked for normality in SAS and analyzed using ANOVA for mixed models using the MIXED procedure of SAS [26] with the diet, time of sampling and their interactions as fixed effects. Anti-KLH IgG data failed to satisfy the Gaussian (normal) distribution, then were ln transformed and rechecked for normality before comparisons were performed. Tukey post-hoc test was used for comparison among groups for cortisol, anti-KLH IgG, and skin test. Comparisons with *p*-value lower than 0.05 were considered statistically significant.

Data of growth, carcass characteristics, pH values, chemical composition, colour and WBSF were subjected to a one-way ANOVA with diet as fixed effect. Individual lamb was the experimental unit. All effects were tested for statistical significance (*p* < 0.05) and significant effects were reported in tables. When significant differences were found (*p* < 0.05), the Student *t*-test was used to locate significant differences between means.

## 3. Results

### 3.1. Metabolic Profile, Immune System and Cortisol Response

The effects of dietary treatments and sampling time on metabolic profile in blood plasma of lambs are shown in Table 2. Lambs supplemented with Q, LS and LS + Q showed lower plasma urea (*p* < 0.001), creatinine (*p* < 0.05), cholesterol (*p* < 0.05), and higher NEFA and triglyceride plasma concentration (*p* < 0.01) than control group. Lambs supplemented with linseed or in combination showed higher plasma levels of total protein (*p* < 0.05) than C and Q groups, while the highest plasma albumin concentration (*p* < 0.01) was found in lambs fed with LS diet respect to all other groups.

In addition, lambs from LS group showed the lowest glucose level (*p* < 0.01), while, lambs supplemented with quinoa seed shows the highest plasma glucose values (*p* < 0.05). GGT enzyme plasma level of LS group was found higher (*p* < 0.001) than all other groups. Referred to sampling time effect, urea (*p* < 0.001), total protein (*p* < 0.05) and ALP (*p* < 0.01) activity showed higher values at the end of trial, while glucose (*p* < 0.05) and cholesterol (*p* < 0.05) values decrease during time with the lowest values at the end of trial.

The effects of dietary treatments and time sampling on antibody titers to KLH are showed in Figure 1.

Anti-KLH IgG production was affected by time of sampling (*p* < 0.001). Each of experimental group registered an increase from starting time, to 15 and to 25 day of the experiment on the anti-KLH IgG produced in plasma lambs. Moreover, an increase between from 15 and to 25 day in all experimental groups was registered.

When cell-mediated immune response of lambs was investigated, a significant effect of time of sampling (*p* < 0.001) and of diet (*p* < 0.01) was found. Q, LS and LS + Q showed an increase from 0 day to 15 day and from 15 to 35 day; whereas, C group displayed a reduction of cell-mediated immune response from 15 to 35 day. At 15 day and 35 day both Q and LS lambs displayed higher skinfold thickness than C and LS + Q lambs (Figure 2).

Plasma cortisol production of loading test is depicted in Figure 3. An increase of cortisol secretion in the blood of lambs after 10 min of loading test in all groups was registered (*p* < 0.001). On average, the LS group displayed the lowest cortisol concentration in comparison to C and Q lambs during the loading test (*p* < 0.001). At 10 min after loading test LS lambs displayed lower cortisol secretion than C; whereas, at 60 min after loading test no differences emerged between groups in cortisol secretion returning to basal level of starting test.

### 3.2. Growth, Carcass Parameters and Meat Quality

The effect of dietary supplementation on growth performance, carcass characteristics and *post-mortem* pH of *quadriceps femoris* muscle is shown in Table 3.

No significant differences in the average daily gain, slaughter weight, carcass weight and dressing percentage were observed among all experimental groups. On the contrary, muscle pH was affected by diet supplementation showing higher values (*p* < 0.05) at 1 and 3 h post-mortem in LS and LS + Q meat compared to Q and control meat. The effect of dietary supplementation with linseed, quinoa seed and their combination on intramuscular fat content, WBSF and colorimetric parameters of *quadriceps femoris* muscle from lambs is reported in Table 4.

Diet had a significant effect on intramuscular fat content (*p* < 0.05) showing the highest values in meat from lambs supplemented with linseed, quinoa seed and their combination. Regarding to organoleptic properties of meat, lambs subjected to dietary supplementation with linseed or/and quinoa seed showed a greater meat tenderness in terms of lower WBSF values (*p* < 0.01) than meat from control group. Dietary supplementation affected the meat color parameters, in particular higher values of lightness, redness and yellowness (*p* < 0.01) were detected in meat from lambs of all supplemented groups.

## 4. Discussion

### 4.1. Metabolic Profile, Immune System and Cortisol Response

Metabolic profile of all experimental groups were in normal range for the young animals of the sheep specie [27,28]. Results highlighted that different feeding strategy can influence the metabolic response of lambs. In the present experiment lambs from all supplemented groups exploited a better efficiency use of nitrogen, kidney functionality and protein metabolism synthesis. Urea levels found in our treated groups are consistent with data reported by Nudda et al. [29] in blood from Saanen goats supplemented with extruded linseed; on the contrary, the same authors did not find any effect of extruded linseed on kidney and liver function, probably due to the different response related to the different age of the animals used in the present study. As regard to lipidic metabolism, the highest NEFA and triglyceride plasma concentration found in all supplemented groups could be related to the high fat content of diets in agreement with Gonthier et al. [30]. Furthermore, supplementation with quinoa and linseed and their combination caused a hypo cholesterol effect in plasma of lambs. This result could be determined by the main constituents of quinoa or linseed (e.g., fiber, flavonoids, polyphenol and protein) that could inhibit absorption of dietary cholesterol through increase cholesterol catabolism in feces [31,32]. This finding confirmed the role of ω-3 enriched seeds to reduce cardiovascular disease risk or to enhance nutritional status in different animal species. Supplementation with linseed or quinoa affected in different manner the glucose level showing an opposite trend. In particular, quinoa seed, rich in carbohydrate and starch led to an higher production of rumen volatile fatty acids or intestinal glucose adsorption with a higher level of glucose in the blood of lambs supplemented with quinoa. On the contrary, the lowest content of non-structural carbohydrates in LS diet led to a lower level of glucose in the blood of lambs supplemented with linseed. 

Cortisol concentration in blood after a stressor, such as transport, increases according to the proportional perception of stress from the animal. In the present experiment, as expected, cortisol levels picked up after 10 min post loading test in all groups returning to basal level when stress was over. Stress response changes according to the type of stress acting and to the animals’ physiological state. In our study the low levels of cortisol secretion in plasma of lambs fed linseed could be responsible for the enhanced cellular immune responses. In a previous study lambs displaying an increased secretion of cortisol during open field test exhibited a reduced cellular response to skin test [33]. In sheep, a relationship between stress, activation of the HPA axis (by assessment of plasma concentration of cortisol), and functions of the immune system was established [34]. In the present study, both low level of cortisol and activation of cell mediated-immune responses found in LS lambs, suggested that diet has a role in the degree of activation of HPA-axis and consequently on activation of innate immune responses. The higher activation of cell-mediated immune response found in Q lambs is of difficult explanation; however, a relation between the high glucose concentrations in blood of Q lambs and the higher cell-mediated response could be hypothesized according to Assmann and Finlay [35] who defined the role of glucose essential as fuel for immune system cells, which require glucose for biosynthesis during their activation to mount an immune response.

A tentative explanation for the absence of effects of linseed supplementation on humoral response of lambs in the present study could be the different nature of experimental procedure as compared to previous experiments by Caroprese et al. [9,10,23]. Furthermore, in the present experiment no acute or chronic stressor was acting on lambs, thus inducing a different humoral immune response as compared with ewes or cows under heat stress in which linseed supplementation caused an increase of anti-OVA IgG concentration during heat stress [9,23].

### 4.2. Growth, Carcass Parameters and Meat Quality

The lack of significant effect on performance in vita among the different dietary treatments could be due to the similar crude protein and metabolizable energy level of diets, in agreement with previous studies [36,37] reporting that carcass traits of lambs were not affected by the type of dietary lipid supplement. In our study, meat from lambs supplemented with linseed, quinoa seed and their combination showed the highest content of intramuscular fat. These results are in agreement with Kott et al. [38], that observed an increase of intramuscular fat deposition of LL in meat of lambs fed with diet supplemented with safflower seed.

Stressful conditions can increase the release of adrenaline leading to an ante mortem glycogen breakdown and, as a consequence, to a lower pH values [39]. Previous studies [40] reported that feeding regimen can protect animals from the potential depletion of glycogen caused by pre-slaughter stressors. In the present study, a minor susceptibility to pre-slaughter stress appears in lambs supplemented with linseed showing the highest pH at 1 and 3 h post-mortem confirming that the high levels in n-3 fatty acids of linseed might have exerted immunomodulatory properties in lambs as suggested by Caroprese et al. [10].

In addition, the major tenderness of meat from lambs supplemented with LS could be due to the highest pH found after slaughtering meaning optimal conditions for the activation of endogenous enzymes responsible for postmortem tenderization. It has been demonstrated that the rate of glycolysis and consequently the rate of pH decline could improve tenderness by altering the optimal conditions of enzyme activity. In the previous study, della Malva et al. [17] highlighted that supplementation with linseed or quinoa was able to influence proteolytic pattern of myofibrillar fraction suggesting that linseed supplementation could be a strategy to improve the tenderness profile of lamb meat.

Color is an important factor determining the consumer purchase of meat [41]. It is known that the lightness, redness and yellowness are affected by several factors, especially the intramuscular fat content, which may alter light scattering properties [42], as well as ultimate pH and feeding system [43]. Our results were in contrast with Moloney et al. [44] that observed a lack of effect of supplementation with unsaturated lipid sources on muscle colour.

The present results confirm that the improvement of animal health can affect lamb’s meat quality in agreement with Napolitano et al. [15].

## 5. Conclusions

This study evaluated the effects of supplementation with linseed, quinoa seed and their combination on metabolic profile, immune system, and cortisol response in blood, and on meat quality of merinos derived lambs. Both linseed and quinoa supplementation enhanced the cell-mediated immune responses of lambs, and showed a hypo cholesterol effect on blood of lambs. Moreover, lambs supplemented with linseed resulted in the lowest level of cortisol secretion after 10 min post loading test. The reduced reactivity to stress by linseed fed lambs could be responsible for the enhanced cellular immune responses highlighting the link between stress and the immune system. This result provides an important contribution to sustain the immune response of lambs. In addition, data from the present experiment highlighted that linseed supplementation in lambs enhance meat quality producing better meat tenderness. These findings should be considered for development of specific strategies aimed at improving the value and quality of meat from merinos derived lambs and sustaining lambs’ welfare.

## Figures and Tables

**Figure 1 animals-08-00204-f001:**
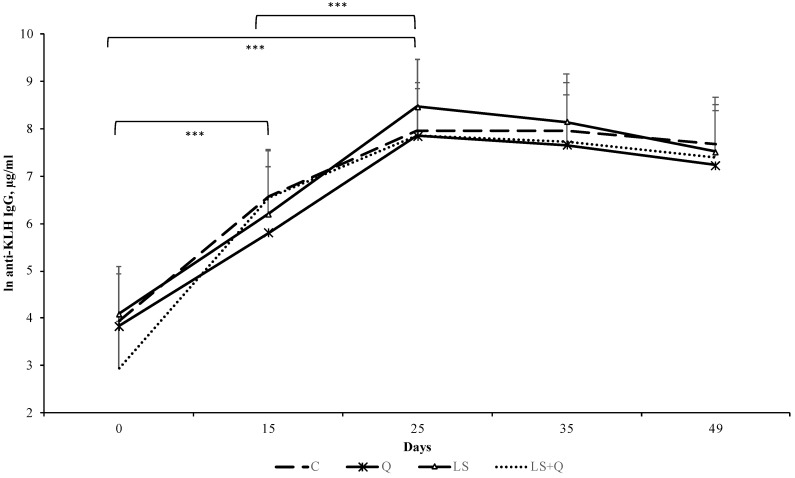
Antibody titers to KLH (Least Squares means ± SEM) detected at 0, 15, 25, 35 and 49 day of the experiment in blood of lambs as affected by different diet supplementation (C = control; Q = quinoa seed; LS = linseed; LS + Q = linseed + quinoa). *** = *p* < 0.001.

**Figure 2 animals-08-00204-f002:**
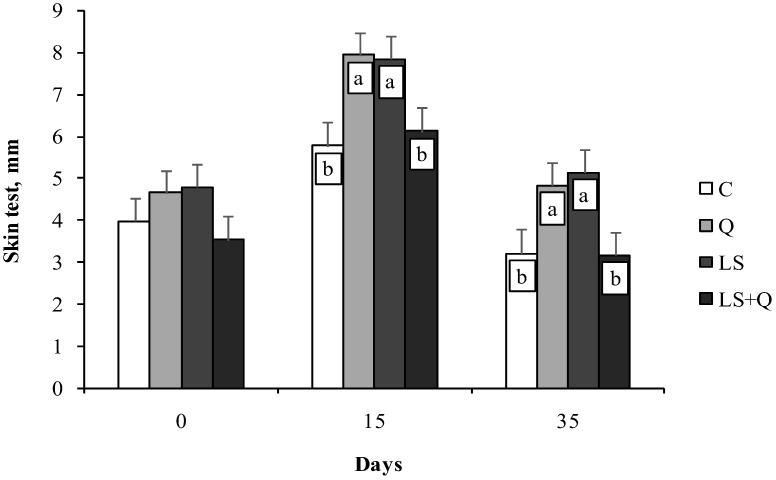
Skinfold thickness (Least Squares means ± SEM) measured after PHA injection at 0, 15, and 35 day of the experiment in lambs as affected by different diet supplementation (C = control; LS = linseed; Q = quinoa seed; LS + Q = linseed + quinoa). ^a,b^ different letters differ among feeding treatments within a sampling day (*p* < 0.05).

**Figure 3 animals-08-00204-f003:**
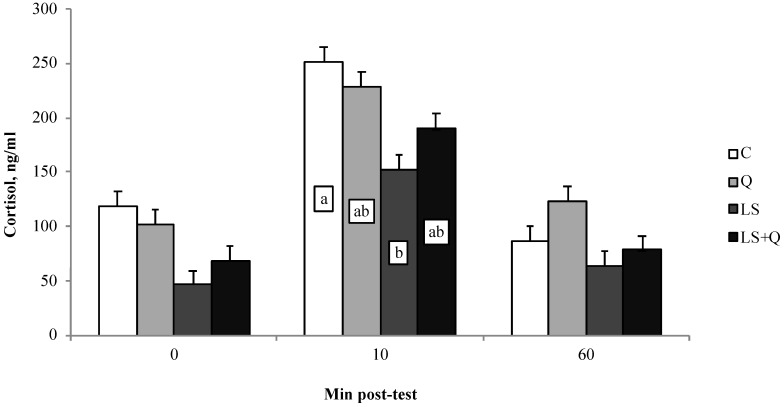
Plasma cortisol production (Least Squares means ± SEM) measured immediately before and then at 10, and 60 min post loading test in lambs as affected by different diet supplementation (C = control; LS = linseed; Q = quinoa seed; LS + Q = linseed + quinoa). ^a,b^ different letters differ among feeding treatments within a sampling day (*p* < 0.05).

**Table 1 animals-08-00204-t001:** Ingredients, chemical and fatty acid composition of experimental diets (C = control; LS = linseed; Q = quinoa seed; LS + Q = linseed + quinoa).

Item	Diet Treatment			
	C	Q	LS	LS + Q
Diet ingredient ^1^, % DM				
Vetch and oat hay	21.18	21.16	21.25	21.13
Concentrate	78.88	67.49	67.32	67.40
Linseed	-	-	11.58	5.79
Quinoa	-	11.35	-	5.67
Chemical composition, % DM				
Crude Protein (CP)	18.75	18.62	19.09	18.86
Ether Extract (EE)	3.34	3.58	6.26	4.92
Neutral Detergent Fiber (NDF)	40.31	41.76	41.75	41.76
Acid Detergent Fiber (ADF)	16.67	15.89	17.69	16.79
Non Structural Carbohydrates (NSC)	28.71	27.81	24.71	26.25
ME (MJ/kg DM)	11.34	11.61	12.12	11.87
Fatty acids composition, %				
C14:0	0.29	0.30	0.17	0.21
C16:0	18.10	18.22	11.03	13.02
C18:0	2.04	2.08	2.54	2.28
C18:1	20.15	20.23	18.55	19.53
C18:2 *c*9,*c*12	51.42	50.99	31.53	38.54
C18:3 *n*-3	5.58	5.66	34.50	24.28

^1^ Ingredients: vit. A, 1150,000 IU; vit. D, 390,000 IU; vit. E, 50 mg; vit. PP, 8500 mg; vit. B1, 112 mg; vit. B2, 112 mg; vit. B6, 80 mg; vit. B12, 1 mg; d-Pantotenic acid, 2400 mg; ChoIine, 15,000 mg; Iron, 150 mg; Manganese, 800 mg; Zinc, 2200 mg; Cobalt, 8 mg; Iodine, 30 mg; Selenium, 5 mg; Molibden, 10 mg.

**Table 2 animals-08-00204-t002:** Effect of diet treatment and time sampling on metabolic profile parameters of merinos-derived lambs (C = control; LS = linseed; Q = quinoa seed; LS + Q = linseed + quinoa seed), (mean ± SEM).

Item		Diet Treatment					Time Sampling (d)						Effects, *p*	
		C	LS	Q	LS + Q	SEM	0	25	35	49	SEM	Diet	Time	Diet × Time
	Urea (mmol/L)	6.20 ^a^	5.16 ^c^	5.71 ^b^	5.61 ^b c^	0.16	5.41 ^b^	5.62 ^b^	5.47 ^b^	6.36 ^a^	0.16	***	***	***
Protein metabolism	Creatinine (µmol/L)	82.08 ^a^	75.42 ^b^	74.59 ^b^	74.86 ^b^	2.15	74.74	75.21	78.38	78.61	2.14	*	NS	NS
	Total protein (g/L)	59.00 ^c^	61.34 ^ab^	58.79 ^c^	60.44 ^bc^	0.53	59.47 ^b^	59.31 ^b^	59.68 ^b^	61.28 ^a^	0.53	**	*	NS
	Albumin (g/L)	36.21 ^bc^	38.34 ^a^	35.34 ^c^	34.25 ^d^	0.37	35.87	35.72	36.11	31.45	0.37	***	NS	NS
	Globulin (g/L)	22.79 ^b^	23.80 ^b^	23.63 ^b^	26.19 ^a^	0.68	23.60	23.60	23.58	24.83	0.68	**	NS	NS
	Glucose (mmol/L)	4.81 ^b^	4.63 ^c^	5.06 ^a^	4.86 ^ab^	0.08	5.14 ^a^	4.91 ^b^	4.75 ^bc^	4.56 ^c^	0.08	**	***	*
	Cholesterol (mmol/L)	1.47 ^a^	1.17 ^b^	1.17 ^b^	1.22 ^b^	0.06	1.54 ^a^	1.37 ^b^	1.23 ^b c^	1.19 ^c^	0.06	***	***	NS
Energy metabolism	Trygliceride (mmol/L)	0.20 ^c^	0.27 ^a^	0.26 ^ab^	0.25 ^b^	0.02	0.26	0.24	0.23	0.23	0.01	**	NS	NS
	NEFA (mmol/L)	0.13 ^b^	0.22 ^a^	0.21 ^a^	0.20 ^a^	0.02	0.20	0.19	0.18	0.18	0.02	**	NS	NS
	ALP (U/L)	235.07	265.84	250.12	245.68	13.38	218.06 ^b^	231.61 ^b^	255.27 ^b^	291.78 ^a^	13.38	NS	**	NS
Blood enzymes activity	LDH (U/L)	972.96 ^b^	1186.15 ^ab^	1279.56 ^a^	1223.91 ^a^	85.4	1138.82	1158.18	1120.95	1244.64	85.40	NS	NS	NS
	GGT (U/L)	71.93 ^b^	83.37 ^a^	72.53 ^b^	74.91 ^b^	2.57	76.51	75.58	75.45	75.19	2.57	***	NS	NS

NS = not significant; * = *p* < 0.05; ** = *p* < 0.01; *** = *p* < 0.001; ^a,b,c^ = values with different letters are significantly different (*p* < 0.05).

**Table 3 animals-08-00204-t003:** Growth performance, carcass characteristics and pH of *quadriceps femoris* from lambs as affected by different diet supplementation (C = control; LS = linseed; Q = quinoa seed; LS + Q = linseed + quinoa) (means ± SEM).

Item	Diet Treatment				SEM	Effect, *p*
	C	Q	LS	LS + Q		
Initial weight (kg)	14.56	14.30	14.43	14.20	0.88	NS
Final weight (kg)	25.41	25.14	24.16	25.69	1.35	NS
ADG (kg/d)	0.23	0.21	0.21	0.23	0.01	NS
Slaughter weight (kg)	24.56	24.74	24.91	24.68	0.15	NS
Carcass weight (kg)	14.41 ^b^	15.03 ^a^	13.98 ^c^	14.62 ^b^	0.21	*
Cold carcass weight (kg)	14.11 ^b^	14.67 ^a^	13.63 ^c^	14.31 ^b^	0.13	*
Dressing percentage (%)	58.63 ^c^	60.75 ^a^	56.11 ^d^	59.25 ^b^	0.16	*
pH 1 h	6.31 ^b^	6.33 ^b^	6.50 ^a^	6.56 ^a^	0.04	*
pH 3 h	6.07 ^b^	6.15 ^b^	6.31 ^a^	6.33 ^a^	0.05	*
pH 6 h	5.88	5.91	6.05	6.00	0.06	NS
pH 24 h	5.73	5.65	5.70	5.67	0.05	NS

NS = not significant; * = *p* < 0.05; ^a,b,c,d^ = Values with different letters are significantly different (*p* < 0.05).

**Table 4 animals-08-00204-t004:** Intramuscular fat content, Warner–Bratzler shear force (WBSF) and color parameters (lightness: L*, redness: a*, yellowness: b*) of *quadriceps femoris* from lambs as affected by different diet supplementation (C = control; LS = linseed; Q = quinoa seed; LS + Q = linseed + quinoa) (means ± SEM).

Item	Diet Treatment				SEM	Effect, *p*
	C	Q	LS	LS + Q		
Intramuscular fat (%)	1.04 ^b^	1.32 ^a^	1.41 ^a^	1.45 ^a^	0.08	*
WBSF (kg/cm^2^)	6.54 ^a^	5.75 ^b^	5.39 ^b^	5.44 ^b^	0.18	**
*L**	38.72 ^b^	45.61 ^a^	44.09 ^a^	44.38 ^a^	0.51	**
*a**	8.75 ^b^	9.67 ^a^	9.52 ^a^	9.57 ^a^	0.25	**
*b**	9.26 ^b^	11.77 ^a^	11.42 ^a^	11.83 ^a^	0.28	**

* = *p* < 0.05; ** = *p* < 0.01; ^a,b^ = Values with different letters are significantly different (*p* < 0.05).

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
