# Peer review of "Effect of Diet Supplementation with Quinoa Seed and/or Linseed on Immune Response, Productivity and Meat Quality in Merinos Derived Lambs"

_animals, 2018, doi:10.3390/ani8110204_

Reviewer 1 Report

Dear authors,

 thank you very much for the well written article entitled “Effect of diet supplementation with quinoa seed and/or linseed on immune response, productivity and meat quality in merinos derived lambs“.

First, I enjoyed the precise description within the whole article. Nevertheless, there are some minor revisions, which are important.

-          Please double-check the formatting of your article, the tables and figures. You have a mix of font sizes (e.g. the description of table 3, the axis label in figure 1 etc.). There are also some unusual line breaks (e.g. table 1). In addition, you have some doubled spaces (e. g. line 69, 95, 239).

-          Abstract: The sentence in line 34 f. “Meat from lambs supplemented with linseed or/and quinoa seed showed higher pH [...], and lower WBSF values than meat from quinoa and control group” could be understood wrong. You mean “. “Meat from lambs supplemented with linseed or/and quinoa seed showed higher pH [...]. Meat from L and LS +Q showed lower WBSF values than meat from quinoa and control group”.

-          Animal welfare includes much more than the described aspect. The word is used excessive recently. You might not overestimate it in your discussion (line 309...) and conclusion.

-          The paragraph line 197-200 is not supported by the figure 2. Please double-check your sentences.

Kind regards

Author Response

Dear authors,

thank you very much for the well written article entitled “Effect of diet supplementation with quinoa seed and/or linseed on immune response, productivity and meat quality in merinos derived lambs“.

First, I enjoyed the precise description within the whole article. Nevertheless, there are some minor revisions, which are important.

Au: We gratefully acknowledge the positive comments and the advices given by this reviewer. The manuscript has been improved according to reviewer’s suggestions.

-   Please double-check the formatting of your article, the tables and figures. You have a mix of font sizes (e.g. the description of table 3, the axis label in figure 1 etc.). There are also some unusual line breaks (e.g. table 1). In addition, you have some doubled spaces (e. g. line 69, 95, 239).

Au: As suggested, the article has been formatted.

-     Abstract: The sentence in line 34 f. “Meat from lambs supplemented with linseed or/and quinoa seed showed higher pH [...], and lower WBSF values than meat from quinoa and control group” could be understood wrong. You mean “. “Meat from lambs supplemented with linseed or/and quinoa seed showed higher pH [...]. Meat from L and LS +Q showed lower WBSF values than meat from quinoa and control group”.

Au: The sentence has been clarified in the text.

-       Animal welfare includes much more than the described aspect. The word is used excessive recently. You might not overestimate it in your discussion (line 309...) and conclusion.

Au: The word “welfare” has been modified in several part of the discussion and conclusion sections.

-      The paragraph line 197-200 is not supported by the figure 2. Please double-check your sentences.

Au: The sentences has been revised according to reviewer’s suggestions.

New and added parts are underlined in red.

Reviewer 2 Report

In this work, the authors evaluate the effects of supplementing the diet of lambs with linseed and quinoa seeds on the welfare, productivity and quality of merino lamb meat.

This contribution is interesting and valuable.

Simply Summary:

14 performances and meat quality of lambs. The relation between nutrition and immune response has been

24 quality producing a better meat tenderness. These findings should be considered for development of specific

 26 Abstract:  In the last years several studies have been investigating/have investigated the strong relation between nutrition and

31 the respective supplementation and control group (C) without supplementation. Lambs from all

36 and lower WBSF values than meat from control group.

2.1. Feed analysis

95 984.13) measurements. Metabolizable energy was calculated using INRA system [20]. Fatty acids

117 of the antigen KLH (Sigma Aldrich, Milan, Italy), to which the animals had not previously been exposed. At

124 with 3% of Bovine Serum Albumin (BSA) dissolved in PBST (200 μl) at 37 °C for 1 h to reduce non-specific

3. Results

3.1. Metabolic profile, immune system and cortisol response

216 displayed lower cortisol secretion than C; whereas, at 60 min after loading test no differences

4.1. Metabolic profile, immune system and cortisol response

238 Metabolic profile of all experimental group was in normal range for the young animals of the

4. Discussion 236

4.1. Metabolic profile, immune system and cortisol response

278 experiment no acute or chronic stressor was acting on lambs, thus inducing a different

279 humoral immune response as compared with ewes or cows under heat stress in which linseed

5. Conclusions

321 lambs enhance meat quality producing better meat tenderness. These findings should be considered for

Author Response

In this work, the authors evaluate the effects of supplementing the diet of lambs with linseed and quinoa seeds on the welfare, productivity and quality of merino lamb meat. This contribution is interesting and valuable.

Simply Summary:

 14 performances and meat quality of lambs. The relation between nutrition and immune response has been

24 quality producing a better meat tenderness. These findings should be considered for development of specific

26  In the last years several studies have been investigating/have investigated the strong relation between nutrition and

 31 the respective supplementation and control group (C) without supplementation. Lambs from all

 36 and lower WBSF values than meat from control group.

95 measurements. Metabolizable energy was calculated using INRA system [20]. Fatty acids

117 of the antigen KLH (Sigma Aldrich, Milan, Italy), to which the animals had not previously been exposed.

124 with 3% of Bovine Serum Albumin (BSA) dissolved in PBST (200 μl) at 37 °C for 1 h to reduce non-specific

216 displayed lower cortisol secretion than C; whereas, at 60 min after loading test no differences

238 Metabolic profile of all experimental group was in normal range for the young animals of the

278 experiment no acute or chronic stressor was acting on lambs, thus inducing a different

279 humoral immune response as compared with ewes or cows under heat stress in which linseed

321 lambs enhance meat quality producing better meat tenderness. These findings should be considered for

AU: We gratefully acknowledge the positive comments and the advices given by this reviewer.

English has been improved in the revised manuscript according to reviewer’s suggestions.

New and added parts are underlined in red.
